# Developments and Applications of Liver-on-a-Chip Technology—Current Status and Future Prospects

**DOI:** 10.3390/biomedicines13061272

**Published:** 2025-05-22

**Authors:** Joseph Mugaanyi, Jing Huang, Jiongze Fang, Arthur Musinguzi, Caide Lu, Zaozao Chen

**Affiliations:** 1Department of Hepato-Pancreato-Biliary Surgery, Ningbo Medical Center Lihuili Hospital, Health Science Center, Ningbo University, Ningbo 315040, China; 2Health Science Center, Ningbo University, Ningbo 315200, China; 3Zuckerberg College of Health Sciences, University of Massachusetts Lowell, Lowell, MA 01854, USA; 4Institute of Biomaterials and Medical Devices, Southeast University, Suzhou 215000, China

**Keywords:** organ-on-chip, liver, microfluidics, liver-on-a-chip

## Abstract

**Background/Objectives**: Liver-on-a-chip (LiOC) technology is increasingly recognized as a transformative platform for modeling liver biology, disease mechanisms, drug metabolism, and toxicity screening. Traditional two-dimensional (2D) in vitro models lack the complexity needed to replicate the liver’s unique microenvironment. This review aims to summarize recent advancements in LiOC systems, emphasizing their potential in biomedical research and translational applications. **Methods**: This narrative review synthesizes findings from key studies on the development and application of LiOC platforms. We explored innovations in material science and bioengineering, including microfluidic design, 3D printing, stem cell– and tissue-derived liver organoid integration, and co-culture strategies. Commercially available LiOC systems and their regulatory relevance were also evaluated. **Results**: LiOC systems have evolved from simple PDMS-based chips to complex, multicellular constructs incorporating hepatocytes, endothelial cells, Kupffer cells, and hepatic stellate cells. Recent studies demonstrate their superior ability to replicate liver-specific architecture and functions. Applications span cancer research, drug toxicity assessment (e.g., drug-induced liver injury prediction with >85% sensitivity), disease modeling, and regenerative medicine. Several platforms have gained FDA recognition and are in active use for preclinical drug testing. **Conclusions**: LiOC technology offers a more physiologically relevant alternative to traditional models and holds promise for reducing reliance on animal studies. While challenges remain, such as vascularization and long-term function, ongoing advancements are paving the way toward clinical and pharmaceutical integration. The technology is poised to play a key role in personalized medicine and next-generation therapeutic development.

## 1. Introduction

Liver pathologies such as hepatocellular carcinoma, fatty liver disease, cholangio-carcinoma, and more are significant contributors to global mortality rates and disease burden [1,2,3,4]. Secondly, the liver is a principal actor in drug metabolism and detoxification, making it an important organ in pharmacodynamics [5,6]. The need to understand the underlying mechanisms of these diseases and to develop better therapeutics necessitates in vitro experimentation. However, for the greater part of modern biomedical research, foundation medical research in vitro has relied on cell-on-a-dish technology. While we have gained tremendous knowledge and insights through studies that relied on the two-dimensional (2D) cell culture, it is not without limitations [7,8]. The most significant limitation is the inability to model the complex nature of organs. Organ-on-chip (OOC) technology has emerged as a revolutionary approach, particularly in modeling and studying complex organ functions and interactions in vitro [9,10]. OOC utilizes microfluidic systems to create biomimetic environments that can replicate the physiological and biochemical characteristics of human and animal organs.

The technology has already been applied in the research of various pathologies in different organs such as skin cancer, kidney diseases, and liver cancer, among others [10,11]. Among the different models that have been developed, the liver-on-a-chip (LiOC) model has gained significant interest in part due to the liver’s critical role in drug metabolism, toxicity assessment, and disease modeling [12,13,14].

Over the past wo decades, numerous studies have demonstrated the potential of LiOC systems to provide insights into hepatic functions and responses to pharmacological agents. For instance, Zhang et al. developed a biodegradable scaffold with built-in vasculature that facilitated the engineering of live tissues, with potential applications in drugs testing and toxicity screening [15]. Similarly, van Midwoud et al. utilized a microfluidic approach to assess interorgan interactions, highlighting the importance of liver function in drug metabolism and the potential for integrating multiple organ systems on a chip [16].

Since the introduction of microfluidics technology, 3D cell cultures, and organ-on-a-chip technology, numerous advancements have been made in the field, especially concerning potential application in live research. While prior reviews have focuses broadly on organ-on-a-chip development, this article emphasis recent advancements in LiOC technologies, with a focus on co-culture strategies, perfusion control innovations, and translational pathways toward regulatory qualification. We aim to synthesize emerging design principles and practical hurdles in the real-world deployment of LiOC platforms.

## 2. Early Developments and Initial Applications

The concept of OOC technology emerged in the early 2000s, with a primary focus on simulating single organ functions [9,17]. The technology builds upon microfluidics technology, encompassing miniaturized chips that contain a single or multiple microchannels ranging from 10–100 µm in diameter [18,19,20,21,22]. The simplest form of a microfluidic is composed of a three-layer sandwich: a top polydimethylsiloxane (PDMS) layer, followed by a porous membrane and bottom PDMS later (Figure 1). With developments in material science techniques and 3D printing and the lowering of fabrication costs, more customized chip designs are possible, allowing for greater control and flexibility when modeling and mimicking different organs [23,24,25]. The journey from 2D cell cultures to 3D organoid-on-chip technology has been multifaceted, including advancements in both material science and bioengineering, from traditional monolayer cultures to spheroids and ultimately organoids on the bioengineering side, and from Petri dishes to microfluidic chips on the material sciences side. The initial models, such as the lung-on-chip developed by Huh et al., demonstrated the potential of microfluidic devices to recreate the mechanical and biochemical environment of human tissues [26,27]. This model allowed for the study of respiratory diseases and the effects of airborne toxins, marking a significant step in the application of OOC technology for disease modeling and drug testing.

### Evolution of Fabrication Techniques

Traditionally, most LiOC devices have been fabricated using PDMS due to its optical transparency, gas permeability, and ease of prototyping [28]. However, PDMS suffers from limitations such as drug adsorption and poor scalability [29]. To address these issues, newer fabrication methods are emerging, particularly 3D printing, digital light processing (DLP), and bioprinting.

Three-dimensional printing and DLP enable rapid, customizable, and scalable chip production with high precision. For example, Wu et al. highlighted the benefits of stereolithography-based 3D printing to create OOC platforms with defined microchannel geometries using biocompatible polymers [30,31]. Studies, such as those by Liu et al., have emphasized the importance of polymer composition, porosity, and flow shear in influencing cell behavior and function in on-chip studies [32,33]. Bioprinting extends this further by integrating cell-laden bioinks, allowing for the spatially controlled deposition of multiple cell types [34,35,36]. Zhang et al. demonstrated a pioneering example using bioprinted iPSC-derived hepatocytes in vascularized liver tissue constructs for on-chip studies [37]. These innovations in fabrication techniques not only enhance physiological fidelity but also improve reproducibility and clinical scalability.

## 3. Hepatic Organoids

A human liver is composed of two basic categories of cells: the parenchymal cells (hepatocytes), which constitute the majority of hepatic cells (80%), and non-parenchymal cells, including resident immune cells (Kupffer cells, KCs), hepatic stellate cells (HSCs), liver sinusoidal endothelial cells (LSECs) and, cholangiocytes [38]. At the foundational level, the cells are organized around a central vein, radiating outwards, forming a polygon bound by connective tissue and with a portal triad at each vertex [39,40]. Each portal triad is composed of a bile duct, a portal vein, and a hepatic artery. This assembly constitutes the hepatic lobule. Liver organoids aim to mimic this composition and structure. There are two potential routes to developing a liver organoid: (1) liver-inducing organoid stem cell culture and (2) liver tissue-derived organoid culture.

### 3.1. Stem Cell-Derived Liver Organoids

Stem cells are progenitors of all other cell types in the body and thus, under proper conditions and stimuli, will differentiate into the desired mature cell type. When considering stem cell-derived organoids, three options are available: (1) adult stem cells (ASCs) [41,42], embryonic stem cell (ESCs) [43], and (3) induced pluripotent stem cells (iPSCs) [44,45,46]. The use of embryonic stem cells is controversial due to ethical concerns. [47] Adult stem cells, sometimes referred to as somatic stem cells, are the resident pluripotent cells scarcely found in native tissues and are responsible for tissue repair in the different organs of the body [48,49]. However, given how scarce they are, they are difficult to extract and study. Adult stem cells typically include hematopoietic stem cells, mesenchymal stem cells, neural stem cells, epithelial stem cells, and skin stem cells. Induced pluripotent stems cells (iPSCs) do not naturally occur in the body; they are laboratory cultured by impregnating a somatic cell with embryonic genes causing the somatic cell to revert to a pseudo-stem cell state. iPSCs are a relatively recent development in stem cell research, only discovered in 2006 [50,51].

The liver typically has no detectable Lgr5 stem cell marker expression levels [52]. However, under conditions of injury, epithelial Lgr5 positive cells are activated to trigger liver regeneration and repair [53]. EpCAM+ bipotent progenitor cells have also been isolated and cultured into organoids [54]. The propagation of the organoid must first have the proper conditions to establish the organoid, then proceeding to a secondary culture with conditions that remove the proliferative signals inhibiting committing to a ductal differentiation and assuming a hepatocyte differentiation. The process of culturing a liver organoid via this route basically involves four steps (Figure 2) as follows:

#### 3.1.1. Isolation and Differentiation of Stem Cells into Hepatic Progenitors

Once the specific type of stem cell to be used has been determined, they are extracted and isolated [55,56]. These human pluripotent stem cells (hPSCs) must then be guided to differentiated into definitive endoderm, a process that is usually achieved by treating the cells with Activin A [57,58]. Once a definitive endoderm is established, further differentiation is induced using growth factors such as fibroblast growth factor 2 (FGF2) and bone morphogenetic protein 4 (BMP4). These factors steer the cells towards a hepatic fate, leading to the generation of hepatic progenitor cells (HPCs) [59].

#### 3.1.2. Maturation into Hepatocyte-like Cells

HPCs are subsequently matured into hepatocyte-like cells (HLCs) by exposure to additional growth factors, including hepatocyte growth factor (HGF) and oncostatin M (OSM) [60,61]. These cells exhibit key characteristics of mature hepatocytes, such as albumin production, urea synthesis, and cytochrome P450 activity, which are critical functions of the liver.

#### 3.1.3. Incorporation of Non-Parenchymal Cells

To closely mimic the native liver microenvironment, LiOC technology integrates non-parenchymal cells into the organoids. LSECs, KCs, HSCs, and cholangiocytes are either co-differentiated with hepatocytes from the same progenitor pool or incorporated through co-culture techniques [62,63]. This multicellular environment is essential for modeling liver-specific interactions and functions, such as immune responses and fibrogenesis.

#### 3.1.4. 3D Culture and Organoid Formation

The differentiated cells are then embedded in a 3D matrix, such as Matrigel, which provides structural support and a scaffold for cell–cell and cell–matrix interactions [64,65,66]. The 3D culture conditions enable the self-organization of cells into hepatic organoids, which exhibit lobule-like architecture with central vein and portal triad-like structures. This spatial organization is crucial for maintaining the polarized nature of hepatocytes (which is lost in 2D cell cultures) and facilitating the exchange of metabolites and signals [67].

### 3.2. Liver Tissue-Derived Liver Organoids

With the prior approach, it is necessary to first isolate the stem cells, whether embryonic or adult. Then, culture the isolated stem cells under the proper stimuli to stimulate their differentiation into hepatic cells. An alternative approach is to culture the liver organoid from a tissue sample of the liver without necessarily isolating the individual cells. With this approach, a tissue sample (liver sample in this case) is obtained either by biopsy or from intraoperative sample collection for patients undergoing hepatic resection. Culturing a liver organoid via this method follows the general steps shown in Figure 2D.

#### 3.2.1. Sample Collection

Obtain a liver biopsy sample through an appropriate method, such as percutaneous or transverse biopsy. The sample must be of adequate size to ensure sufficient yield [68,69]. The sample is stored in a tissue preservation medium at 4 °C during transportation. The sample must be processed within than 24 h of collection.

#### 3.2.2. Tissue Preparation and Cell Isolation

The sample is cleaned and cut into small fragments. The sample is then enzymatically digested to isolate hepatocytes and non-parenchymal cells [70].

#### 3.2.3. Cell Expansion

The isolated liver cells are expanded in vitro under specific conditions that promote their proliferation using growth factors like HGF and FGF2. Over the first week, primary liver cells (PLCs) will proliferate and form a halo. Continue to maintain the culture under optimal conditions, typically in a humidified incubator at 37 °C [68].

#### 3.2.4. Sub-Culture and Maintenance

Once sufficient PLCs are present, subculture them into larger dishes or high-throughput culture dishes to expand the cell population. Regularly change the culture medium and monitor for cell health and differentiation markers [70].

#### 3.2.5. Organoid Formation

To induce organoid formation, transfer the cultured cells into a suitable 3D matrix or scaffold that mimics the liver environment. This can involve extracellular matrix components like Matrigel or collagen to support three-dimensional growth

Whether the organoid is cultured via the stem cell route or from biopsy tissue, once the liver organoid has been attained, its functionality must be assessed by evaluating liver-specific functions, such as protein synthesis. This can involve biochemical assays to measure enzyme activity and other liver-specific markers. A comparison of liver organoid culture from hPSCs versus from liver biopsy samples is summarized in Table 1.

### 3.3. Tumor-Derived Liver Organoids

Tumor-derived liver organoids, also known as tumoroids, represent a special kind of liver organoid specifically intended for the study of liver cancer biology and personalized therapeutics. These organoids are cultivated from primary tumor tissues, retaining the histological architecture, genetic stability, and functional characteristics of the original tumors [74]. Research has demonstrated that liver organoids derived from various types of liver cancer, such as hepatocellular carcinoma and cholangiocarcinoma, can effectively mimic the heterogeneity and complexity of the tumors from which they are derived [75,76]. For instance, studies have shown that they preserve the tumor’s genomic landscape and can differentiate between various tumor subtypes even after prolonged culture, making them invaluable for understanding tumor evolution and treatment responses [77,78,78].

The utility of tumor-derived liver organoid extends beyond basic research; they serve as powerful platforms for drug screening and biomarker identification. For example, organoids derived from patient samples have been used to test the efficacy of therapeutic agents, leading to the identification of potential treatments such as the ERK inhibitor SCH772984 for primary liver cancer [75,76,79]. Furthermore, tumoroids can be co-cultured with stromal and immune components to recreate the tumor microenvironment, facilitating a more comprehensive understanding of liver cancer progression, immune involvement, and responses to targeted therapies.

## 4. Liver-on-a-Chip

In 2010, Huh et al. from Harvard’s Ingber laboratory published their paper “Reconstituting Organ-Level Lung Functions on a chip” in *Science* [26], in which they detailed their approach to designing biomimetic microsystem that could mimic the functional alveolar-capillary interface of the human lung. Building on this technology, they developed a liver on chip that replicated liver tissue architecture, blood flow, and cellular interactions. Their design consists of parallel channels lined with various cells from the liver such as hepatocytes, liver sinusoidal endothelial cells, and hepatic stellate cells. The chip uses a microfluidic system to simulate blood flow and can recreate species-specific toxicity responses to drugs, which can be insightful when predicting how different compounds will affect human health compared to animal models. Studies have shown that the Ingbar Liver Chip can achieve high sensitivity (87%) and specificity (100%) in predicting drug-induced liver injury, outperforming traditional animal models [80]. The chip can be used to model liver diseases, providing insights into disease mechanisms and potential therapeutic targets. The US FDA has utilized data from these chips to evaluate drug safety, indicating their growing acceptance in regulatory frameworks.

Another advantage of the LiOC compared to traditional animal models is the ability to conduct high-throughput studies, which can be further enhanced by big data analysis. Lensing Taylor’s lab focuses on integrating LiOC devices with sophisticated data analysis and computational modeling to stimulate and study liver functions, particularly for drug metabolism and liver disease research. Liu et al. proposed a trivascular liver-on-a-chip (TVLOC) device composed of a hepatic artery, a portal vein, and a central vein [81]. They coupled this with a bilayer microsphere approach to yield a chip with bilayer microspheres of different cell types, which they co-cultured. Their LiOC design was shown to replicate the substance concentration gradient seen in the liver microenvironment. Various research teams are developing distinct models to replicate liver function, and LiOC technology is evolving rapidly. Table 2 summarizes a comparison of some of the major researchers in the field, highlighting the cellular components and tissue structures utilized in their liver chips.

From a design perspective, most LiOC platforms fall into two categories: linear microchannels (for example, Ingber Lab’s) and zonated or gradient-based architectures (such as McCarty et al.) [26,86]. While linear designs enable easier fabrication and scaling, they lack the physiologic gradients found in native liver lobules. Gradient systems, although complex, better replicated zonal metabolism and enzyme distribution. With respect to co-culture strategies, layered systems like SQL-SAL offer better compartmentalization but may hinder direct cell–cell communication. On the other hand, mixed cultures facilitate interaction but complicate readouts and maintenance.

### Diversity of Liver-on-a-Chip Models

A number of implementations of LiOC platforms exist, each tailored for different research goals. Organoid-on-chip systems combine self-assembling 3D liver organoids with microfluific flow to replicate liver physiology [89]. These systems are ideal for modeling development and cell signaling, but may lack architectural precision [90]. Bioprinted liver constructs, on the other hand, use extrusion or inkjet printing to spatially position hepatocytes and stromal cells within ECM-like scaffolds [91,92,93]. Their advantage over basic OOC systems is the potential for defined geometry and the ability to better mimic tissue architecture.

Tumor-on-chip models integrate patient-derived tumoroids with endothelial or stromal components, recreating the tumor microenvironment for personalized drug testing [94,95,96]. Perfused multicellular systems use vascular-mimicking channels lined with hepatocytes, stallate, and endothelial cells, providing high fidelity for metabolism, fibrosis, or immune interactions.

## 5. Applications of Liver Organoids and Liver-on-a-Chip

### 5.1. Cancer Research

LiOC has become a pivotal tool in liver cancer research, offering a sophisticated platform to study the complex interactions within the liver microenvironment and the behavior of cancer cells (Figure 3). Liver cancer, particularly hepatocellular carcinoma, remains a significant health challenge due to the complexity of tumor heterogeneity and metastasis. Tumoroids derived from cancerous liver tissue offer a way to model the tumor biology [97,98,99]. The modeled tumoroids can then provide insights into tumor progression and therapeutic responses [100]. By mimicking the tumor microenvironment, researchers are able to study how liver cancer cells interact with their surroundings, facilitating the identification of novel therapeutic targets. They also serve as a platform for testing anti-cancer drugs and personalized therapies [96]. Advances like organoid-based CRISPR technologies and 3D bioprinting further enhance the modeling of liver tumors, potentially improving the models and the downstream translational benefits [101,102,103].

### 5.2. Drug Development and Toxicity Testing

One of the primary applications of LiOC systems is in drug development, particularly in assessing drug-induced liver injury (DILI), a leading cause of drug clinical drug withdrawal. A recent study demonstrated that a human LiOC could accurately predict hepatotoxicity for a range of drugs, achieving a sensitivity of 87% and a specificity of 100% in identifying known hepatotoxic compounds [80]. This capability could significantly reduce the rate of drug attrition in clinical trials, potentially saving the pharmaceutical industry significant funds annually by improving the predictive power of preclinical models. LiOC systems have also been shown to model diverse mechanisms of toxicity and to measure relevant clinical biomarkers, outperforming animal models in predicting human responses to drugs [104,105]. Several well-characterized compounds have been tested using LiOC platforms to validate their predictive capabilities. For example, acetaminophen, a prototypical hepatotoxin, consistently causes dose-dependent cytotoxitiy and glutathione in LiOC models, mimicking human clinical heaptotoxicty profiles [106,107].

Troglitazone, withdrawn due to liver failure risk, induces mitochondrial damage and bile acid accumulation in dynamic LiOC systems, which are not observed in standard 2D cultures [97,108]. Other studies have shown that a microfluidic model could predict idiosyncratic DILI for drugs like diclofenac and nefazodine, demonstrating > 85% concordance with clinical outcomes [109,110]. These works support the use of LiOC in early-phase safety screening and mechanistic toxicology. Figure 4 highlights a schema for LiOC application in drug toxicity analysis.

### 5.3. Biological Mechanism Research

Increasingly, liver organoid-on-a-chip systems have proven to be valuable tools for studying biological mechanisms in the liver. For instance, studies have used the liver-on-a-chip technology to investigate the RAS-RAF-MAPK and P13K-PTEN/AKT pathways which are involved in EGFR activation [111]. While these pathways have previously been studied in animal models, the LiOC platform enables researchers to manipulate the pathway parameters in a controlled environment that mimics the in vivo liver microenvironment. Since the LiOC incorporated microfluidic systems, they allow for the precise control of fluid flow, shear stress, mechanical forces, and chemical gradients in the organoid. Studies have successfully investigated the effects of shear stress on hepatocyte function and drug metabolism [87]. LiOCs also allow for the integration and co-culture of multiple cell types and tissue systems. Thus, studies have utilized the platform to study the interactions between different liver cell types and how these interactions influence signaling pathways involved in liver fibrosis and regeneration [14]. Figure 5 demonstrates a schema of LiOC utilization in biological mechanism studies.

Furthermore, a key advantage of LiOC for studying biological mechanisms is the ability to perform real-time, high-resolution imaging and analysis. With this capability, researchers are able to observe dynamic changes in signaling pathways and cell behavior over extended periods [14,87]. For instance, fluorescent and live cell imaging have been used to track the activation of key signaling molecules, such as NF-κB and Wnt, in response to various stimuli within LiOC [112].

### 5.4. Disease Modeling

LiOC systems also play an increasing role in modeling liver diseases, including hepatitis, fibrosis, fatty liver disease, and cirrhosis. By introducing disease-causing mutations or exposing organoids to injury-inducing substances like free fatty acids, researchers are able to study the progression of these conditions at a molecular level [113,114]. Furthermore, by incorporating patient-derived cells, the LiOC can allow for the exploration of liver disease mechanisms in a controlled environment that closely mimics the in vivo conditions. LiOCs have been used to study the pathology of non-alcoholic fatty liver disease (NAFLD) and its progression to fibrosis and cirrhosis [115,116]. They are also being used to study hepatitis as well as liver fibrosis. This approach has the added benefit of enabling researchers to investigate the pathophysiology of liver diseases and simultaneously test potential therapeutic interventions in a more relevant context. Figure 6 highlights key aspects of LiOC applications in biomedical research.

### 5.5. Regenerative Medicine

The liver is perhaps the organ in the body with the highest regenerative potential. However, many of the mechanisms behind this capability are still not well understood [117,118]. LiOCs derived from iPSCs hold promise for exploring and gaining deeper insight into the mechanism of liver regeneration. Beyond that, there is also potential for cell-based therapies and liver transplantation. Patient cell-derived organoids could potentially replace damaged tissue in patients with chronic liver diseases. Recent studies have demonstrated the ability of liver organoids to integrate with host tissue and perform essential liver functions in animal models, suggesting a future where bioengineered liver tissues could be used in human patients [119,120,121]. LiOC systems have enabled the maturation and functional validation of stem cell-derived hepatocytes prior to in vivo use [122,123]. For example, Calabrese et al. cultured iPSC-derived hepatocytes on a perfused chip for 14 days before transplanting them into Fah^−/−^ mice, resulting in the partial restoration of albumin secretion and ammonia detoxification [68]. Similarly, recent studies have demonstrated that human hepatocyte organoids, expanded in vitro, can engraft into injured murine livers and proliferate extensively, mimicking the liver’s regenerative response. Furthermore, advancements in microfluidic technologies have enabled the vascularization of hepatic organoids in vitro, providing a platform for studying liver tissue engineering and disease modeling [124,125]. However, challenges like the vascularization and long-term survival of transplanted organoids remain an enigma yet to be addressed [126].

### 5.6. Commercialization and Regulatory Relevance

Over the past half decade, several LiOC platforms have transitioned from academic prototypes to commercial systems. Emulate Inc.’s (Boston, MA, USA) Liver Chip uses polystyrene-based channels with a co-culture of hepatocytes and liver sinusoidal endothelial cells and is validated by the FDA under its Innovatice Science and Technology Approaches for New Drugs (ISTAND) program. Ewart el la. showed that LiOC platforms can play a critical role in the optimization of toxicology screening and preclinical assessment [80]. LiOC platforms are already helping address the challenges in drug development that arise from the poor predictive validity of preclinical models that is reflected in the high attrition rate in human trials. CN Bio’s PhysioMimix™ system, (Cambridge, UK) on the other hand, incorporates a single-well open-flow system that is ideal for long-term culture and high-content imaging. They introduced the system in 2021 as a next-generation platform combining in vitro 3D liver models with a range of other organs. Such an advancement opens up avenues for recapitulating multi-organ and systemic effects seen in vivo. Since the liver’s function is tightly linked to other organs like the kidney, having the ability to design and combine multi-organs on a chip can be invaluable to translational medical research. Lim et la. in their 2023 study found that PhysioMimix demonstrated the highest metabolic function with primary human hepatocytes regardless of co-culture with THP-1/kupffer cells for up to 14 days compared to multi-well plate cultures [127]. While most LiOC systems rely on either a pumping mechanism or on rockers to perfuse the LiOC, Mimetes’ OrganoPlate^®^ (Gaithersburg, MD, USA) uses phaseguide-based perfusion. The design utilizes gravity to drive unidirectional fluid flow through the chip’s microfluidic channels. Furthermore, the platform has a membrane-free layered setup which mimicks physiological conditions in which the liver sinusoidal layers are not separated by a memebrane from the hepatocytes and HSCs. Bircsak et al. demonstrated and validated an automated high-throughput hepatotoxicity screening platform composed of iPSC-derived hepatocytes in ECM in a co-culture with LSECs and THP-1-derived macrophages using Mimetas OrganoPlate 2-lane. In their study, the established LiOC remained stable and viable for 15 days [128]. These platforms vary in throughput, media handling, and biomarker integration. With the FDA and other regulators around the world looking to encourage the adoption of organ-on-chip systems and similar technologies to replace animal experiments, more platforms are bound to be introduced to bridge the need. Table 3 summaries these difference and the extent of their regulatory partnerships.

## 6. Future Prospects

Looking ahead, LiOC systems still face several key challenges. First, maintaining long-term hepatic function beyond 28 days remains difficult due to cell de-differentiation and the loss of polarization. Primary hepatocytes often lose polarity and CYP450 expression within 7–14 days under static culture. While perfusion extends viability, many chips still show significant decline in albumin production or urea synthesis beyond 21 days. Furthermore, CYP3A4 and CYP1A2 activity, critical for drug metabolism, decreases by over 50% after 2 weeks without co-culture [129,130]. Second, robust vascularization and stable perfusion channels are complicated to model and are often lacking, which can compromise nutrient and oxygen gradients. The integration of real-time biosensors (such as TEER, lactate, and oxygen) remains rare but is potentially essential for drug screening. Bubble formation, inconsistent flow, and channel occlusion continue to limit reproducibility in long-term experiments [131,132]. Lastly, the lack of standardized protocols hampers reproducibility across labs, which is a prerequisite for regular adoption. For example, iPSC-derived hepatocytes vary by donor line and differentiation protocol, complicating standardization [133]. Addressing these issues will require collaboration between engineers, biologists, and regulators.

(1)Scalability and High-Throughput Screening: Scalability is a critical priority for the future of LiOC to meet the needs of the pharmaceutical industry and research institutions. Current LiOC systems are often limited by the number of replicates that can be run simultaneously. This significantly restricts their utility in large-scale drug screening applications. Efforts are underway to automate LiOC production and operation, allowing for high-throughput screening [134,135].(2)Multi-Organ-on-chip integration: The development of multi-organ-on-chip systems that integrate the LiOC with other organs such as the kidney are yet another promising avenue that could provide a more comprehensive view of drug effects, disease progression, and organ interactions [136]. Multi-organ systems on a chip could also be vital to studying systemic diseases like metabolic syndromes, multi-organ failure, and metastasis. Advancements in microfluidics and 3D printing could potentially provide a breakthrough in this aspect [137].(3)Personalized Medicine and Precision Therapy: The medical research community is increasingly exploring the adoption of a more personalized approach to medical care, especially with regards to therapeutics. Since LiOC can be developed using patient-derived cells, individualized liver models are possible [138]. With these, researchers and clinicians could predict how an individual will respond to a given medical intervention. In the future, it is conceivable that each patient could have their own LiOC to test therapeutic options, provided that the challenges of scalability and cost are addressed.(4)Artificial Intelligence and Machine Learning: As LiOC models become more sophisticated, machine learning model and artificial intelligence (AI) integration are likely to enhance the capabilities of models. The AI algorithms could be employed to analyze the complex data generated by the LiOC systems, identifying patterns and predicting outcomes that may not be immediately apparent to researchers and clinicians [139]. For instance, AI could be utilized to optimize the conditions for organoid growth, predict drug responses, or identify biomarkers associated with specific liver diseases. They could also help accelerate the design of new LiOC platforms by predicting how different cell types, structures, and environmental conditions may affect organ phenotype and function.(5)Addressing Current Challenges in Vascularization: Currently, the vascularization of LiOCs and indeed organoids of other organs, like the lung and kidney, remain a significant challenge [140]. For LiOCs, their inability to replicate the liver’s intricate vascular network, which is essential for nutrient delivery, waste removal, drug metabolism, and plays a role in liver regeneration, is a critical limitation. Perhaps developments in the field may enable creating functional vasculatures within the organoid. Bioprinting technologies are being explored as a potential solution, but it is still early to tell.(6)Clinical Translation: Ultimately, the most exciting prospect for LiOC technology lies in its potential clinical applications, particularly in regenerative medicine. LiOC systems may be used to bioengineer liver tissue for transplantation in patients with liver failure and chronic liver disease [141]. This would help address the shortage of donated livers and would also negate the rejection of the graft. However, this remains a long-term prospect.

## 7. Conclusions

In conclusion, liver-on-a-chip (LiOC) systems have emerged as a transformative tool in liver research, providing more physiologically relevant models for studying liver diseases, drug development, and toxicity testing compared to traditional 2D cultures. They could also potentially reduce our reliance on animal models in both medical research and drug development, thus alleviating the ethical concerns associated with animal experimentation. With advances in stem cell biology, bioengineering, and microfluidics, LiOC systems have the potential to bridge the gap between in vitro experimentation and clinical application. However, challenges such as improving vascularization, maintaining long-term functionality, and scaling for broader clinical use remain. Continued interdisciplinary collaboration will be crucial in refining this technology and unlocking its full potential in personalized medicine and regenerative therapy.

## Figures and Tables

**Figure 1 biomedicines-13-01272-f001:**
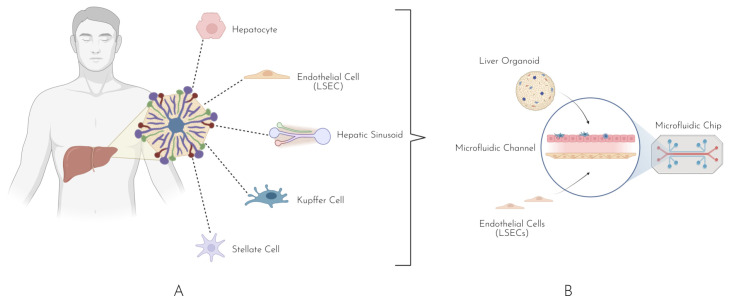
(**A**) Foundational building block of the liver, the hepatic lobule composed of hepatocytes, liver sinusoidal endothelial cells, Kupffer cells, hepatic stellate cells, and the vasculature. (**B**) The microfluidic chip models basic hepatic tissue integrating the liver organoid and endothelial cells on the chip with the microfluid channel modeling vasculature.

**Figure 2 biomedicines-13-01272-f002:**
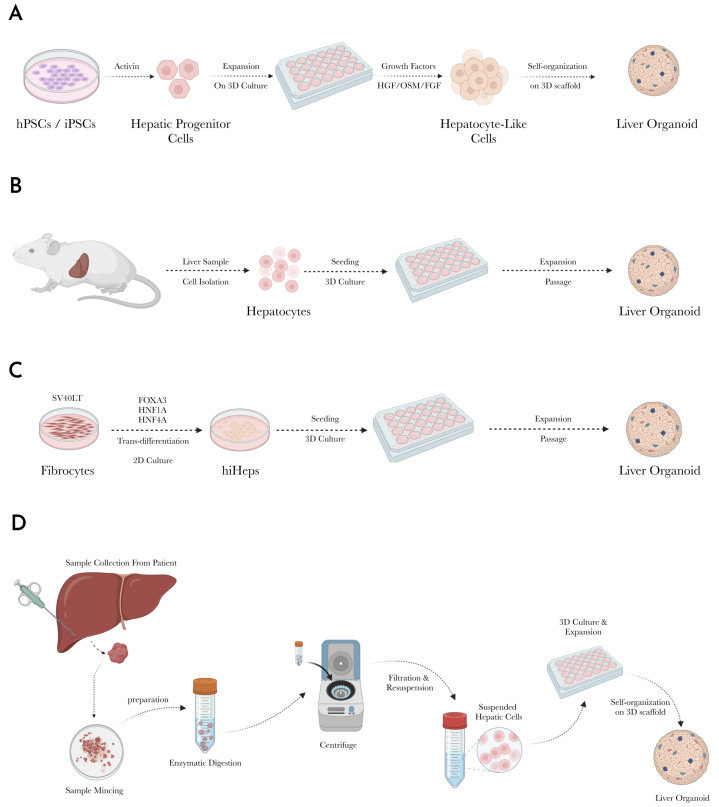
(**A**) Schematic diagram of culture of liver organoid from induced pluripotent stem cells (iPSC). (**B**) Schematic diagram of the culture of the liver organoid from primary liver tissue sample. (**C**) Generation of liver organoid from trans-differentiated fibroblasts. (**D**) An overview schematic diagram of steps to culture a patient-derived liver organoid.

**Figure 3 biomedicines-13-01272-f003:**
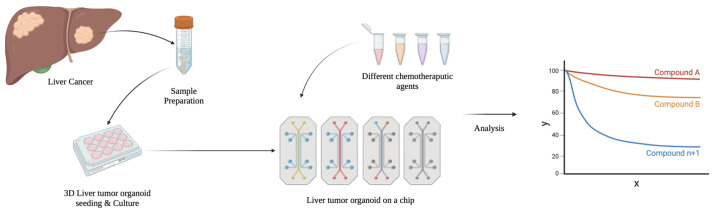
Application of biomimetic liver-on-a-chip (LiOC) platform in liver cancer research. A sample of the tumor is obtained from the patient and cultured into a tumor organoid which can be used to investigate the response to treatment on a microfluidic chip among other potential investigations.

**Figure 4 biomedicines-13-01272-f004:**
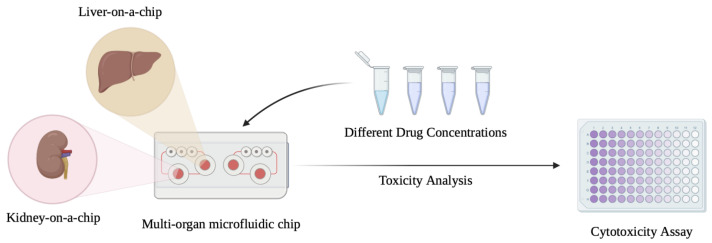
Application of liver-on-a-chip (LiOC) platform in drug development and toxicity analysis. Single-organ microfluidic chips like the LiOC and multi-organ chips are being used to study and develop new therapeutics, as well as to assess their toxicity.

**Figure 5 biomedicines-13-01272-f005:**
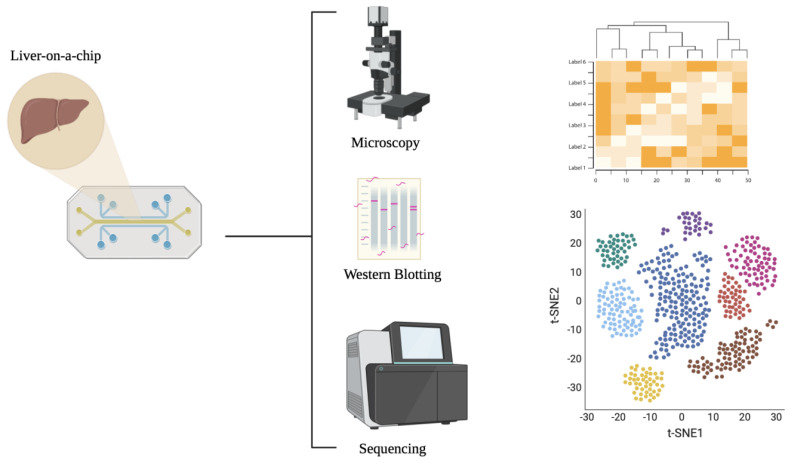
Application of liver-on-a-chip (LiOC) platform in biological mechanisms studies. High-throughput liver-on-a-chip systems are being used to study biological mechanisms in the liver and liver pathologies.

**Figure 6 biomedicines-13-01272-f006:**
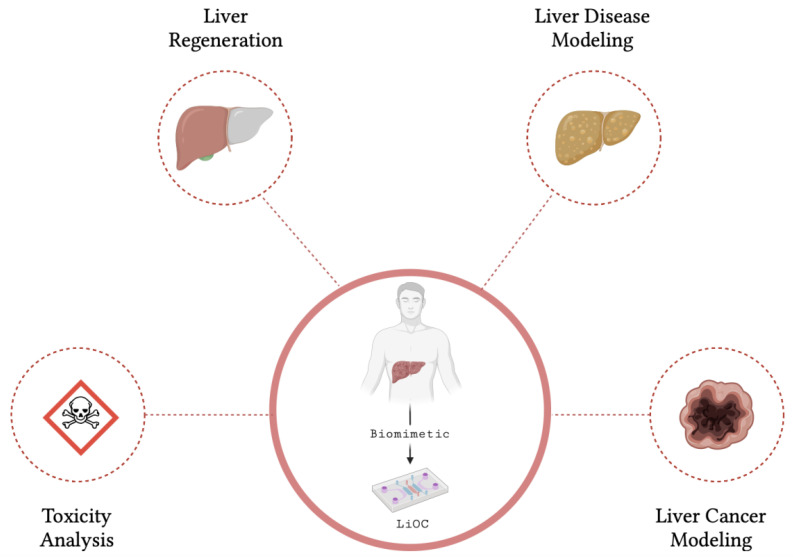
Application of biomimetic liver-on-a-chip (LiOC) platform in biomedical research. LiOCs are being applied across four core aspects of biomedical research: liver cancer modeling and therapeutics; liver disease modeling; regenerative medicine; and drug toxicity research.

**Table 1 biomedicines-13-01272-t001:** Liver organoid culture from hPSCs versus from liver biopsy sample.

Aspect	Source of Culture Cells	
	hPSCs	Biopsy Sample
Cell Source	Derived from human pluripotent cells (hPSCs), including ESCs and iPSCs [71]	Derived from PLCs obtained from biopsy samples [68]
Availability	Unlimited supply as hPSCs can be expanded indefinitely [71,72]	Limited supply due to the small size of biopsy samples, which restricts the number of cells available for culture [73]
Differentiation Process	Requires specific differentiation protocol to mimic liver development stages, involving growth factors and specific media [71]	Involves spontaneous outgrowth of PLCs from biopsy tissue which may not allow for selective cell type enrichment [68]
Time to Generate Organoids	Longer, as it involves several stages of differentiation and maturation (weeks)	Shorter, as liver cells are already committed to a liver lineage (days to a few weeks)
Growth Factors Required	Sequential use of Activin A, FGF2, BMP4, HGF, and oncostatin M to drive differentiation	Limited growth factors needed; typically used factors like HGF and Wnt for cell expansion and maturation
Culture Conditions	Maintained in a defined growth medium supplemented with various growth factors and hormones [71]	Cultures in a medium that supports the outgrowth of PLCs, typically using a reduced growth factor matrix
Cell Composition	Can be engineered to reflect specific cell types and functions, achieving a more controlled composition [71,72]	May not fully recapitulate the diverse cell types present in the original tissue due to outgrowth limitations [68]
Complexity of Organoids	Incorporate hepatocytes and non-parenchymal cells but may need additional co-culture systems for complete cell representation	Naturally contain all liver cell types, including non-parenchymal cells, leading to higher initial complexity
Functional Assessment	Potentially better at mimicking liver function due to controlled differentiation and composition	May exhibit variable liver functions depending on the success of PLC outgrowth and the inherent characteristics of the biopsy sample
Applications	Primarily used for drug screening, disease modeling, and regenerative medicine due to their versatility	Primarily utilized for studying liver pathologies and personalized medicine, but with limitations in functional modeling
Ethical Considerations	Ethical concerns may arise with the use of ESCs; iPSC generation avoids many of these issues	Minimal ethical concerns, as the organoids are derived from patient biopsy tissue
Challenges	Requires careful optimization of differentiation protocols and culture conditions to maintain functionality	Faces challenges such as low cell yield, genetic heterogeneity, and difficulty in maintaining the native microenvironment

**Table 2 biomedicines-13-01272-t002:** Summary of liver-on-a-chip designs from different research teams.

Research Group/Model	Cell Types Included	Key Features
Ingber Lab (Wyss Institute) [26]	Human Hepatocytes, Endothelial Cells, Kupffer Cells, Stellate Cells	Derived from PLCs Microfluidic channels; species-specific toxicity modeling
University of Pittsburgh [82]	Hepatocytes, Stellate Cells, Kupffer Cells, En-dothelial Cells	Self-assembling plate-like structures; fluo-rescent biosensors
Columbia University [83]	iPSC-Derived Hepatocytes, Endothelial Cells	3D biomaterial environment; integration with other tissue types
University of Birmingham [84]	Liver Blood Vessel Cells, Immune Cells	Real-time tracking of immune cell behavior; immunotherapy focus
MIT and Boston University [85]	Primary Hepatocytes, Endothelial Cells	Multi-organ platform for integrated drug screening
McCarty et al. [86]	Primary Human/Rat Hepatocytes	Gradient generator for zonal metabolic studies
Tri-Vascular Liver-on-a-Chip (TVLOC) [81]	Hepatocytes, HSCs, LSECs, KCs	Trivascular system; substance concentration gradient; PMMA microchannels
SQL-SAL [87]	Hepatocytes, Endothelial Cells, KCs, HSCs	Sequentially layered self-assembly liver model.
vLAMPS [88]	Primary Hepatocytes, LSECs, HSCs, KCs	Oxygen gradient replication, functional acinar modeling

**Table 3 biomedicines-13-01272-t003:** Summary of select commercially available liver-on-a-chip platforms.

Company	Platform Name	Key Features	Regulatory Partnerships	Translational Highlights
Emulate (Boston, MA, USA)	Human Emulation System™ (Liver-Chip S1)	PDMS-based microfluidic chip with co-culture of hepatocytes and non-parenchymal cells	(1) CRADA with FDA for toxicology research; (2) accepted into FDA’s ISTAND pilot program for DILI prediction	(1) Partnered with Janssen and Takeda; (2) demonstrated >85% DILI prediction accuracy
CN Bio (Cambridge, UK)	PhysioMimix^®^ Liver MPS	(1) Perfused 3D liver model with open-well design; (2) compatible with imaging and long-term culture	(1) Extended collaboration with FDA’s CDER; (2) used in FDA preclinical workflows	(1) Supported IND-enabling studies for INI-822 (NASH therapeutic); (2) data used in regulatory documentation
Mimetas (Leiden, The Netherlands)	OrganoPlate^®^	96-well microfluidic platform without pumps; ideal for high-throughput screening	Collaboration with HUB for disease model development	(1) Widely adopted in pharmaceutical R&D; (2) used in toxicology and fibrosis modeling
Organovo (San Diego, CA, USA)	ExVive™ Human Liver Tissue	Bioprinted 3D liver constructs from primary human cells	No formal regulatory partnerships reported	Applied in preclinical DILI and fibrosis studies
InSphero (Schlieren, Switzerland)	3D InSight™ Human Liver Microtissues	Multicellular 3D spheroids in hanging-drop or plate format	No formal regulatory partnerships reported	Used extensively in drug discovery pipelines

## Data Availability

Data are contained within the article.

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
