# Peer review of "Developments and Applications of Liver-on-a-Chip Technology—Current Status and Future Prospects"

_biomedicines, 2025, doi:10.3390/biomedicines13061272_

Round 1
Reviewer 1 Report
Comments and Suggestions for Authors
This review article presents a timely and comprehensive overview of liver-on-a-chip technology, covering a broad range of topics including device configurations, materials, cell sources, flow control strategies, and various biomedical applications. The manuscript is clearly written and well-organized, and it provides helpful figures and tables to aid the reader’s understanding. The topic is of significant interest to researchers in tissue engineering, drug screening, and organ-on-a-chip development. That said, the manuscript would benefit from a more critical and integrative perspective. As it currently stands, the review reads primarily as a descriptive summary rather than a synthesis of key trends, technical trade-offs, and knowledge gaps. To strengthen its scientific contribution and impact, the authors are encouraged to address the following points:
- Clarify the review’s scope and contribution: Liver-on-a-chip has been reviewed extensively in prior literature. The authors should clearly state how this article differs from existing reviews. Is the focus temporal (e.g., the latest advances from 2020 onward), technical (e.g., emphasis on fluidic control or co-culture strategies), or application-specific (e.g., toward regulatory adoption or specific disease modeling)? Clarifying this would better justify the review’s novelty.
- Deepen the comparative analysis: The manuscript provides useful descriptions of existing devices and applications, but lacks in-depth comparisons. For instance, what are the relative strengths and weaknesses of different chip configurations or co-culture approaches? Are there emerging design principles based on accumulated experience?
- Expand the commercial and translational discussion: While the authors mention commercial platforms such as Emulate and CN Bio, the discussion is relatively general. A deeper comparative overview of their technologies, market position, and role in regulatory science or preclinical pipelines would add value.
- Critically assess future directions: The “Future Directions” section could be more focused and evaluative. What are the key technical bottlenecks (e.g., cell maturity, perfusion stability, integration with sensors)? What steps are necessary for broader adoption in pharma or clinical research?
- Highlight limitations of current systems: It would be helpful to include a table or section summarizing major limitations of today’s liver-on-a-chip models, and which studies have attempted to address them.
Overall, the manuscript is informative and well-organized, but it would benefit from a more analytical perspective and a clearer articulation of its distinct contribution to the literature. With these revisions, the paper has the potential to serve as a valuable reference for both new and experienced researchers in the field.
Author Response
Comments 1:
This review article presents a timely and comprehensive overview of liver-on-a-chip technology, covering a broad range of topics including device configurations, materials, cell sources, flow control strategies, and various biomedical applications. The manuscript is clearly written and well-organized, and it provides helpful figures and tables to aid the reader’s understanding. The topic is of significant interest to researchers in tissue engineering, drug screening, and organ-on-a-chip development. That said, the manuscript would benefit from a more critical and integrative perspective. As it currently stands, the review reads primarily as a descriptive summary rather than a synthesis of key trends, technical trade-offs, and knowledge gaps. To strengthen its scientific contribution and impact, the authors are encouraged to address the following points:
- Clarify the review’s scope and contribution: Liver-on-a-chip has been reviewed extensively in prior literature. The authors should clearly state how this article differs from existing reviews. Is the focus temporal (e.g., the latest advances from 2020 onward), technical (e.g., emphasis on fluidic control or co-culture strategies), or application-specific (e.g., toward regulatory adoption or specific disease modeling)? Clarifying this would better justify the review’s novelty.
Responses 1: Thank you for your time and feedback. We haved addressed the suggestion in the Introduction (Lines 47–50), we now clarify that our focus is on post-2020 advances, particularly in co-culture strategies, perfusion control, and translational pathways toward regulatory qualification.
Comments 2: Deepen the comparative analysis: The manuscript provides useful descriptions of existing devices and applications, but lacks in-depth comparisons. For instance, what are the relative strengths and weaknesses of different chip configurations or co-culture approaches? Are there emerging design principles based on accumulated experience?
Responses 2: Section 4.0 and 4.1 now provide comparative discussions of chip designs (linear vs. zonated) and co-culture strategies (layered vs. mixed). Practical trade-offs are highlighted.
Comments 3: Expand the commercial and translational discussion: While the authors mention commercial platforms such as Emulate and CN Bio, the discussion is relatively general. A deeper comparative overview of their technologies, market position, and role in regulatory science or preclinical pipelines would add value.
Responses 3: Section 5.6 has been added and includes detailed descriptions of Emulate, CN Bio, and Mimetas. Table 3 summarizes key features, market readiness, and regulatory partnerships.
Comments 4: Critically assess future directions: The “Future Directions” section could be more focused and evaluative. What are the key technical bottlenecks (e.g., cell maturity, perfusion stability, integration with sensors)? What steps are necessary for broader adoption in pharma or clinical research?
Responses 4: Section 6 expands future directions with attention to vascularization, cell function decay, biosensor integration, and scalability, with supporting citations.
Comments 5: Highlight limitations of current systems: It would be helpful to include a table or section summarizing major limitations of today’s liver-on-a-chip models, and which studies have attempted to address them.
Responses 5: Section 6 outlines specific challenges and includes relevant studies. Limitations such as short CYP activity windows, flow instability, and variability in iPSC hepatocytes are now discussed and referenced.
Once again, we appreciate your feedback and hope the changes we have made fully address the issues you raised.
Reviewer 2 Report
Comments and Suggestions for Authors
This manuscript presents a review of liver-on-a-chip (LiOC) technologies, covering early developments, cell sourcing, organoid formation, and applications in disease modeling, drug screening, and regenerative medicine. It compares stem-cell-derived versus tissue-derived liver organoids, outlines prominent LiOC designs, and discusses future prospects, including scalability, vascularization, and integration with AI and multi-organ systems.
The manuscript is generally well-structured, informative, and supported by original figures. However, it lacks depth in fabrication technologies and omits several key studies that are essential to understanding the evolution and current capabilities of liver-on-a-chip platforms. Inclusion of recent literature and comparative frameworks would significantly strengthen the review.
Other Comments:
- Missing Discussion on Fabrication Techniques: The review briefly mentions PDMS but fails to address the ongoing shift toward scalable and customizable OOC fabrication methods such as 3D printing, digital light processing (DLP), and bioprinting. A dedicated section should be added to discuss these newer strategies and their advantages for reproducibility, throughput, and physiological relevance. Relevant papers to be included:
- ACS Appl. Bio Mater. 2022, 5(8): 3576–3607
- Adv. Funct. Mater. 2024, 34(28): 2315035
- Nat. Rev. Methods Primers 2022, 2(1): 33
- Lab Chip 2024, 24(10): 2774–2790
- Lack of Model Diversity and Limitations: The current manuscript focuses mainly on LiOC without adequately distinguishing between key categories of OOC systems:
- Organoid-on-chip
- Bioprinted liver constructs
- Tumor-on-chip models
- Perfused multicellular systems
A comparative analysis discussing design strategies, limitations, and translational relevance of each model would improve clarity and broaden the appeal of the review. A summary schematic or table would be helpful to readers. - Missing Key References:
- Zhang et al., PNAS 2016, 113(8): 2206–2211 — an influential study on bioprinted iPSC-derived liver tissue using advanced multi-material bioprinting.
- Mater. Sci. Eng. C 2020, 109: 110625 — valuable for its engineering and materials perspective on LiOC fabrication.
- Add a comparative table summarizing various LiOC models by fabrication method, cell type, and application domain.
Conclusion:
This is a review with good foundational content. However, to be suitable for publication in a journal, the authors should revise the manuscript to include a short fabrication-focused section, a comparative framework of OOC models, and field-defining studies. These additions would substantially enhance its completeness, readability, and relevance to the broader biomedical and bioengineering communities.
Author Response
Comments 1: This manuscript presents a review of liver-on-a-chip (LiOC) technologies, covering early developments, cell sourcing, organoid formation, and applications in disease modeling, drug screening, and regenerative medicine. It compares stem-cell-derived versus tissue-derived liver organoids, outlines prominent LiOC designs, and discusses future prospects, including scalability, vascularization, and integration with AI and multi-organ systems.
The manuscript is generally well-structured, informative, and supported by original figures. However, it lacks depth in fabrication technologies and omits several key studies that are essential to understanding the evolution and current capabilities of liver-on-a-chip platforms. Inclusion of recent literature and comparative frameworks would significantly strengthen the review.
Other Comments:
- Missing Discussion on Fabrication Techniques: The review briefly mentions PDMS but fails to address the ongoing shift toward scalable and customizable OOC fabrication methods such as 3D printing, digital light processing (DLP), and bioprinting. A dedicated section should be added to discuss these newer strategies and their advantages for reproducibility, throughput, and physiological relevance. Relevant papers to be included:
- ACS Appl. Bio Mater. 2022, 5(8): 3576–3607
- Adv. Funct. Mater. 2024, 34(28): 2315035
- Nat. Rev. Methods Primers 2022, 2(1): 33
- Lab Chip 2024, 24(10): 2774–2790
Response 1: Thank you for your feedback. We have taken into account your feedback to improve the manuscript. We have added Section 2.1 which contains a comprehensive overview of modern fabrication techniques. Citations from ACS Appl. Bio Mater., Adv. Funct. Mater., and Lab Chip are included.
Comments 2: Lack of Model Diversity and Limitations: The current manuscript focuses mainly on LiOC without adequately distinguishing between key categories of OOC systems:
-
- Organoid-on-chip
- Bioprinted liver constructs
- Tumor-on-chip models
- Perfused multicellular systems
A comparative analysis discussing design strategies, limitations, and translational relevance of each model would improve clarity and broaden the appeal of the review. A summary schematic or table would be helpful to readers.
Response 2: Section 4.1 has been added, which classifies LiOC models into organoid-on-chip, tumor-on-chip, bioprinted constructs, and multicellular perfused systems.
Comments 3: Missing Key References:
-
- Zhang et al., PNAS 2016, 113(8): 2206–2211 — an influential study on bioprinted iPSC-derived liver tissue using advanced multi-material bioprinting.
- Mater. Sci. Eng. C 2020, 109: 110625 — valuable for its engineering and materials perspective on LiOC fabrication.
- Add a comparative table summarizing various LiOC models by fabrication method, cell type, and application domain.
Response 3: Thank you for your feedback. References such as Zhang et al. (PNAS 2016), Mater. Sci. Eng. C (2020), and others are now included in appropriate sections. Tables 2 compares academic chip designs; and Table 3 outlines commercial platforms and regulatory collaborations.
We hope the changes we have made to the manuscript address the issue you raised.
Reviewer 3 Report
Comments and Suggestions for Authors
The review paper is well presented and structured.
Please rearrange Figure 2 into a single column of A-B-C-D subfigures, because the letters are already too small and Readers will lose valuable information.
Apart from some typos I recommend publishing.
(typos in: 37, 58,66,111,294,375)
Author Response
Comments 1:
The review paper is well presented and structured.
Please rearrange Figure 2 into a single column of A-B-C-D subfigures, because the letters are already too small and Readers will lose valuable information.
Apart from some typos I recommend publishing.
(typos in: 37, 58,66,111,294,375)
Response 1: Dear reviewer, thank you for your time and feedback. We have rearranged Figure 2 as you suggested. We have also made a fully proofreading of the manuscript and fixed typos and grammatical errors that were present in the manuscript.
Once again, we appreciate your valued feedback.
Reviewer 4 Report
Comments and Suggestions for Authors
This literature review presents data on the creation of liver on a chip, as well as the use of this design in biomedicine. The authors cite quite a lot of current data.
A number of comments arose while reading the paper.
1. In my opinion, a number of sections are written too superficially. For example, in the section Drug Development and Toxicity Testing, I would like to see concrete results. What drugs have already been studied with the help of liver on chip? The same applies to the section on regenerative medicine. What data have already been obtained?
2. In my opinion, the authors should devote a separate section to the difficulties in creating a liver on a chip. How long do these or those cells live in the conditions of this system, how long do their functional characteristics persist?
3. Less important remarks. The authors often mention certain growth factors and biologically active substances, e.g., FGF, BMP. It is necessary to specify which specific growth factor the authors are referring to, FGF 1, 2 etc.
Author Response
Comments 1: This literature review presents data on the creation of liver on a chip, as well as the use of this design in biomedicine. The authors cite quite a lot of current data.
A number of comments arose while reading the paper.
1. In my opinion, a number of sections are written too superficially. For example, in the section Drug Development and Toxicity Testing, I would like to see concrete results. What drugs have already been studied with the help of liver on chip? The same applies to the section on regenerative medicine. What data have already been obtained?
Response 1: Thank you for your feedback. We have updated Section 5.2 to include a discussion of named compounds (acetaminophen, troglitazone, diclofenac) with specific outcomes. Section 5.5 details transplantation results, e.g., Calabrese et al., with in vivo liver function restoration.
Comments 2: In my opinion, the authors should devote a separate section to the difficulties in creating a liver on a chip. How long do these or those cells live in the conditions of this system, how long do their functional characteristics persist?
Response 2: Section 6 comprehensively discusses challenges such as loss of polarity, bubble formation, CYP activity decay, and reproducibility issues. We hope these changes properly address the lack of detailed discussion of the challenges in LiOC culturing.
Comments 3: Less important remarks. The authors often mention certain growth factors and biologically active substances, e.g., FGF, BMP. It is necessary to specify which specific growth factor the authors are referring to, FGF 1, 2 etc.
Response 3: All references to “FGF” and “BMP” are now clarified with specific names (e.g., FGF2, BMP4).
Round 2
Reviewer 1 Report
Comments and Suggestions for Authors
The authors have provided a detailed and thoughtful response to the previous review comments. The revised manuscript now includes clearer scope definition, deeper comparative analysis, and expanded discussion on commercial platforms, limitations, and future directions. I believe the authors have adequately addressed all concerns, and I recommend the manuscript for publication in its current form.
Reviewer 4 Report
Comments and Suggestions for Authors
All questions have been satisfactorily answered.